

# Phytoplankton calcifiers control nitrate cycling and the pace of transition in warming icehouse and cooling greenhouse climates

Karin F. Kvale[1], Katherine E. Turner[1,2], Angela Landolfi[1], and Katrin J. Meissner[3,4]

[1]GEOMAR Helmholtz Centre for Ocean Research, West shore campus, Duesternbrooker Way 20, D-24105 Kiel, Germany
[2]Dept. of Earth, Ocean, and Ecological Sciences, Univ. of Liverpool, Nicholson Building, 4 Brownlow St., Liverpool Merseyside, L69 3GP, UK
[3]Climate Change Research Centre, Level 4 Mathews Building, UNSW, Sydney, NSW, Australia
[4]ARC Centre of Excellence for Climate Extremes

**Correspondence:** K.F. Kvale (kkvale@geomar.de)

**Abstract.** Phytoplankton calcifiers contribute to global carbon cycling through their dual formation of calcium carbonate and particulate organic carbon (POC). The carbonate might provide an efficient export pathway for the associated POC to the deep ocean, reducing the particles' exposure to biological degradation in the upper ocean and increasing the particle settling rate. Previous work has suggested ballasting of POC by carbonate might increase in a warming climate, in spite of increasing

carbonate dissolution rates, because calcifiers benefit from the widespread nutrient limitation arising from stratification. We compare the biogeochemical responses of three models containing 1) a single mixed phytoplankton class, 2) additional explicit small phytoplankton and calcifiers, and 3) additional explicit small phytoplankton and calcifiers with a prognostic carbonate ballast model, to two rapid changes in atmospheric $CO_2$. The first $CO_2$ scenario represents a rapid (150 year) transition from a stable icehouse climate (285 ppm) into a greenhouse climate (1257 ppm); the second represents a symmetric rapid

transition from a stable greenhouse climate into an icehouse climate. We identify a slope change in the global net primary production response with a transition point at about 3.5°C global mean sea surface temperature change in all models, driven by a combination of physics and biology. We also find that in both warming and cooling scenarios, the application of a prognostic carbonate ballast model moderates changes in carbon export production, suboxic volume, and nitrate sources and sinks, reducing the long-term model response to about one-third that of the calcifier model without ballast. Explicit small

phytoplankton and calcifiers, and carbonate ballasting, increase the physical separation of nitrate sources and sinks through a combination of phytoplankton competition and lengthened remineralization profile, resulting in a significantly higher global nitrate inventory in this model compared to the single phytoplankton type model (15% and 32% higher, for icehouse and greenhouse climates). Higher nitrate inventory alleviates nitrate limitation, increasing phytoplankton sensitivity to changes in physical limitation factors (light and temperature). This larger sensitivity to physical forcing produces stronger shifts in ocean

phosphate storage between icehouse and greenhouse climates. The greenhouse climate is found to hold phosphate and nitrate deeper in the ocean, despite a shorter remineralization length scale than the icehouse climate, because of the longer residence times of the deep water masses. We conclude the global biogeochemical impact of calcifiers extends beyond their role in global carbon cycling, and that the ecological composition of the global ocean can affect how ocean biogeochemistry responds to climate forcing.





# 1 Introduction

The importance of pelagic calcifiers to marine carbon export production is well-established Balch (recently summarized by 2018). Pelagic calcifiers contribute about 10% of the total particulate organic carbon (POC) export out of the global euphotic

zone (Jin et al., 2006). Their production of calcium carbonate contributes an additional 1.08-1.60 Pg C y$^{-1}$ as particulate inorganic carbon (PIC; Balch, 2018). This estimate is only 1-3% of the total annual POC export (Balch, 2018), but a global average carbonate content of 34% in ocean sediments (Archer, 1996) suggests a disproportionately large role of PIC for deep-sea carbon export.

It has been proposed that sinking PIC provides protection against microbial remineralization to associated POC, and in-

creases its sinking rate (Armstrong et al., 2002). This "ballast hypothesis" is supported by satellite and sediment trap analyses that suggest an increase in organic carbon transfer efficiency occurs in the presence of PIC (Francois et al., 2002), and by a tight correlation between POC and PIC found in the traps (Armstrong et al., 2002; Klaas and Archer, 2002). Up to 83% of global POC export below 1000 m water depth is associated with PIC (Klaas and Archer, 2002). However, the exact mechanisms that couple inorganic and organic carbon particle export and their impact on ballasting remains debated (discussed in the

recent paper by Rosengard et al., 2015). Wilson et al. (2012) failed to find a robust and globally uniform relationship between sediment trap PIC and POC when taking into account spatial variability in their samples. Passow and De La Rocha (2006) took a mechanistic approach and demonstrated that inclusion of minerals in organic carbon aggregates decreased the particles' volume and porosity, but caused the particles to fragment at high mineral concentrations. They suggested a complex relationship between mineral content and organic carbon export and proposed that POC in fact ballasts PIC (Passow and De La Rocha,

2006). However, a more recent study of sampled carbon fluxes by Rosengard et al. (2015) generally supports the PIC-ballasts-POC hypothesis. Whatever the coupling mechanism, it is possible to represent a robust correlation between PIC and POC deep ocean export in ocean biogeochemical models using the simple Armstrong et al. (2002) ballasting framework. This has been done by Kvale et al. (2015a), who introduced a model of explicit small phytoplankton and calcifiers, and prognostic calcium carbonate to the University of Victoria Earth System Climate Model (UVic ESCM; Weaver et al. 2001). This model associates

a fixed fraction of calculated POC export to a fraction of PIC export, increases the associated POC particles' sinking rate, and stops remineralization (Armstrong et al., 2002). Quasi-thermodynamic dissolution of PIC "releases" a proportionate amount of ballasted POC, slowing its sinking rate and starting remineralization (Kvale et al., 2015a). The net effect is a lengthened remineralization profile for PIC-associated POC, which is supported by sediment trap data (e.g., Rosengard et al., 2015). This ballast scheme significantly improves the representation of ocean nitrate in the UVic ESCM (Kvale et al., 2015a) with respect

to World Ocean Atlas nitrate (Garcia et al., 2009), for reasons discussed in the following sections.

Early work on ocean acidification suggested the ballast effect is at risk from ocean acidification weakening phytoplankton calcification rates (Riebesell et al., 2000), however these concerns have since been reduced by studies demonstrating net increases in calcification and primary production with increasing pCO$_2$ (Iglesias-Rodriguez et al., 2008), adapatability of



phytoplankton calcifiers to acidified conditions (Lohbeck et al., 2012; Liu et al., 2018), adaptability of primary producer assemblages to acidification (Hoppe et al., 2018), and field evidence of historical increases in calcification over the past 200 years (Iglesias-Rodriguez et al., 2008). Looking into the geological record, coccolithophores were more robust (Bolton et al., 2016) and more ubiquitous in warmer, more acidic oceans of the past (Hannisdal et al., 2012).

Recent observed expansion of coccolithophore habitat (Neukermans et al., 2018) agrees with ecological modelling of explicit phytoplankton calcifiers in a warming (Kvale et al., 2015b) and more acidified environment (Furukawa et al., 2018). Coccolithophores have lower nutrient requirements than many other phytoplankton species, which increases their relative fitness as circulation slows and the surface ocean stratifies (Kvale et al., 2015b). There may be a benefit to their expansion over hundred-year timescales in a warming future; a modelling study suggests carbonate ballasting has a stabilizing effect on export

production, reducing the long-term decline in deep ocean export relative to a model where ballasting is not included, even when export losses due to ocean acidification are considered (Kvale et al., 2015b). This stabilization of export production occurs because an increasing proportion of the exported POC is protected from near-surface remineralization by PIC, which slows the transition into a more "heterotrophic" export regime (relatively larger primary production and near-surface export values than modern, but smaller deep ocean export fluxes). Continued carbon and nutrient export to the deep ocean damps the response of

ocean primary production and remineralization to near-surface warming by removing nutrients from fast upper-ocean recycling (Kvale et al., 2015b).

A damped increase in remineralization, which consumes oxygen, should also slow the expansion of suboxic volume in a warming ocean. This mechanism linking PIC ballast to ocean suboxia was identified previously by Hofmann and Schellnhuber (2009) in the context of projected losses of PIC ballast due to ocean acidification increasing remineralization rates and suboxic

volume. Kvale et al. (2015b) found the opposite effect by PIC ballasting on changes in ocean suboxia in a warming climate because in their model, ocean acidification did not significantly reduce PIC export.

Hofmann and Schellnhuber (2009) described an ocean suboxic volume control by PIC ballasting that has the potential to change in a changing climate. We describe an additional and previously unrecognized control by phytoplankton calcifiers on ocean biogeochemistry that also arises from PIC ballasting. By stabilizing suboxic volume, calcifiers mitigate changes in

global ocean nitrogen fixation and denitrification rates during rapid changes in climate. Also in our model, competition with other phytoplankton types and the longer remineralization length scale due to PIC ballasting increases global nitrate inventory by spatially separating nitrogen fixation from denitrification (described by Landolfi et al., 2013, as the "vicious N cycle"). This effect increases in warmer climates in which calcifiers benefit from more nutrient-limited conditions and compose a greater proportion of the total phytoplankton biomass. In contrast to primary production, export production and suboxia, ocean

phosphate storage becomes more sensitive to rapid changes in climate when calcifiers and PIC ballasting are included explicitly. Larger shifts in phosphate storage occur because all phytoplankton are less nitrate-limited, and are therefore more sensitive to changes in physical limitation factors (light and temperature). We explore this control in two sets of idealized experiments- a rapid icehouse to greenhouse transition, and a rapid greenhouse to icehouse transition. The physical responses of the ocean circulation to the atmospheric $CO_2$ forcing are described in Kvale et al. (2018).




## 2 Methods

For our experiment we use the University of Victoria Earth System Climate Model (UVic ESCM) version 2.9 (Weaver et al., 2001; Meissner et al., 2003; Eby et al., 2009; Schmittner et al., 2005, 2008). The experimental set-up is identical to that of Kvale et al. (2018, and described below); the current study is an expansion of the ramp-up (icehouse to greenhouse) and ramp-

down (greenhouse to icehouse) analyses documented there to include the ocean biogeochemical model outputs from the same data files. We employ the same three biogeochemical model configurations (MIXED, CAL, and NOCACO3TR) used in Kvale et al. (2015b, also described below).

Our climate model contains an ocean model, land and vegetation components, dynamic-thermodynamic sea ice model, and sediments. The atmosphere is represented by a two-dimensional energy-moisture balance model. Winds are prescribed

from monthly NCAR/NCEP reanalysis data and are adjusted geostrophically to surface pressure changes calculated based on temperature anomalies (Weaver et al., 2001). Continental ice sheets are fixed to a modern configuration (Weaver et al., 2001). The horizontal resolution is $3.6°$ longitude $\times$ $1.8°$ latitude, and there are 19 vertical levels in the ocean.

The standard UVic ESCM version 2.9 includes a NPZD submodel with "mixed phytoplankton" and "diazotroph" phytoplankton functional types and one "zooplankton" functional type (Schmittner et al., 2008). The MIXED model configuration

is the standard UVic ESCM NPZD submodel, updated with the changes made by Keller et al. (2012). The CAL model configuration builds upon MIXED and additionally includes a "small and calcifying phytoplankton" functional type (simplified to "calcifiers" throughout this manuscript) and a prognostic $CaCO_3$ tracer (Kvale et al., 2015a). The NOCACO3TR model configuration only includes the additional "small and calcifying phytoplankton" functional type and does not include the prognostic $CaCO_3$ tracer. This intermediary model is used to distinguish between model behavior arising due to phytoplankton compe-

tition from that arising due to the use of the prognostic $CaCO_3$ tracer. Table 1 summarizes relevant biogeochemical model tracers and functional types for the three model configurations used here. Additional model details can be found in Kvale et al. (2015a, b).

We first integrate the 3 model configurations (MIXED, NOCACO3TR, and CAL) for more than 20,000 years in 2 climate configurations to achieve 6 equilibrium climate states. The first climate configuration, for the ramp-up simulations (hereafter,

RU), equilibrates the biogeochemical models with an atmospheric $CO_2$ concentration of 285 ppm and represents an icehouse climate. The second, for the ramp-down simulations (hereafter, RD), equilibrates the biogeochemical models with an atmospheric $CO_2$ concentration of 1257 ppm to represent a greenhouse climate. Solar and orbital forcing are prescribed at modern levels in all configurations.

We then force each respective climate configuration with a 1% per year increase (for RU) or 1% per year decrease (for

RD) in atmospheric $CO_2$ concentration for 150 years to reach the other experiment's initial $CO_2$ concentration (Figure 1). Each model is then integrated an additional 350 years, keeping atmospheric $CO_2$ concentrations fixed at year 150 levels. No non-$CO_2$ greenhouse gas forcings are included in the simulations.





## 3 Simulation results

As in Kvale et al. (2015b), the physical response of the three biogeochemical model configurations to each set of radiative forcings are nearly identical. The CAL model RD displays a slight difference in physical response relative to the other two (Figure 1) due to a significantly longer spin-up (100 KY as opposed to 20 KY), in which internal drift resulted in small

differences in initial state. In the RU, global ocean ventilation shoals and maximum northern hemisphere overturning declines from about 20 Sv at year 1 to about 11 Sv at integration year 400 (Kvale et al., 2018). Global average surface air temperature increases by more than 6.8°C, and zonally averaged upper ocean temperatures increase by as much as 8.6°C by year 500 (Kvale et al., 2018). Antarctic Bottom Water formation collapses (Kvale et al., 2018). In RD, the greenhouse ocean starts with an initial average temperature nearly 6°C warmer than the icehouse ocean, and the greenhouse atmosphere is nearly 7°C

warmer (Kvale et al., 2018). Meridional overturning accelerates rapidly in both hemispheres in RD, with maximum northern hemisphere overturning exceeding 40 Sv by year 150, and maximum southern hemisphere overturning strengthening to nearly 20 Sv (Kvale et al., 2018). Stonger overturning is accompanied by a dramatic deepening and expansion of ventilation sites in the Southern Ocean, North Pacific, and North Atlantic (Kvale et al., 2018).

### 3.1 Calcifiers and ballasting mitigate changes in global carbon export

Large changes occur in carbon export (Figure 2) in all models in RU and RD, as the models transition between more "autotrophic" and more "heterotrophic" export regimes (Kvale et al., 2015b). The more autotrophic export regime is the initial RU state and final RD state, and is qualitatively described as a regime in which moderate amounts of POC and PIC are produced and exported from the surface, and remineralize/dissolve in the deep ocean. The more heterotrophic export regime is the inital RD and final RU state, in which larger amounts of POC and PIC are produced in the upper ocean, but less POC survives

upper-ocean remineralization to reach the deep ocean. Figure 2 presents change in POC and PIC for each model, where each model has unique absolute POC and PIC export values (therefore we do not assign specific POC and PIC export values to categorize export regimes). Additional information is supplied as an appendix (Appendix A) that provides more insight into these regime transitions. In RU, early small declines in near-surface POC export (around 0.1 Pg C y$^{-1}$) occur in all models due to declining net primary production (NPP) prior to the transition into a more heterotrophic regime (see Appendix A for

a description of NPP responses). Around year 170, near-surface POC export starts to increase in models that do not include ballasting as near-surface stratification starts to reduce (not shown), SSTs continue to warm, and heterotrophic rates continue to increase. Alleviated stratification, as well as increased nutrient recycling, drives an increase in NPP (and POC) in these models (Kvale et al., 2015b). However, the inclusion of PIC ballast in CAL causes near-surface POC export not to increase because the expanding calcifier population provides protection to an increasing proportion of the total POC from near-surface reminer-

alization (Fig. 2 and Kvale et al., 2015b). Protection of POC from near-surface remineralization by PIC reduces the amount of particle breakdown in the near surface, which reduces the availability of dissolved nutrients available for new primary production, thus mitigating a potential increase in near-surface POC export as the model becomes increasingly heterotrophic. Deep ocean POC export does not recover from early declines, and remains lower than the initial value for the duration of the RU





simulation in all models, with an average export decline of 0.016 Pg C y$^{-1}$ at year 500. PIC export drops in the near surface and in the deep ocean in CAL and MIXED as a response to initial declines in NPP (the NPP of calcifiers in CAL, and the NPP of mixed phytoplankton in MIXED), and then recovers as NPP increases in all models. A mitigated PIC export increase in CAL compared to the other two models (i.e., less than one-third of the NOCACO3TR response) is due to a combination of

a mitigated NPP increase and increasing PIC dissolution rates with increasing ocean acidification. The magnified response in PIC export in NOCACO3TR is a result of the magnified NPP response in this model, with calcifiers responding favorably to stratification.

    In RD, MIXED and CAL models show increases in POC and PIC export at all depths prior to the transition to the more autotrophic export regime (Fig. 2). This is driven by the temporary increase in NPP in both models, which is a result of the

upward mixing and advection of nutrients (see acceleration of overturning in Figure 1). Interestingly, NOCACO3TR experiences a decline in NPP over this period but an increase in POC export. An increase in POC export concurrent with a decline in NPP reflects decreasing respiration rates. Why this occurs only in NOCACO3TR becomes apparent by examining the PIC export plots, which show a decline in PIC at all depths in this model only. In the model, PIC export production is calculated as a fixed fraction of POC export production from calcifiers and zooplankton only. Therefore a decline in PIC can occur as a

response to a loss of calcifiers and zooplankton, but still accompany an increase in POC produced by mixed phytoplankton and diazotrophs. Early losses of calcifiers result in a magnified response in the NOCACO3TR model (again, a response in shallow PIC more than 3x larger than that of CAL by year 500, and a more rapid decline in shallow POC than that in CAL over the last 250 years of the simulation). Over the last 250 years of the simulations, all models simulate declines in shallow POC and shallow and deep PIC as NPP declines and they transition into the more autotrophic export regime. Deep ocean POC export

increases as cooling temperatures reduce respiration rates.

    In both RU and RD a mitigating effect on the transition between export regimes (the last 250 years of the simulations) is apparent in the CAL model. In RU this mitigation is due to the increasing ballast effect, which reduces the response of near-surface POC export, and surface and deep ocean PIC export, to less than one-third that of NOCACO3TR. In RD this mitigation is partly due to the loss of calcifiers and consequent declining ballast effect in the high latitudes (Fig. A3), which

leaves an increasing proportion of POC exposed to near-surface respiration, thus reducing the global rate of decline of NPP and POC export. Also, a mutually-beneficial relationship between diazotrophs and calcifiers establishes in the low latitudes in RD. Calcifiers ballast carbon and nutrients away from the surface. Denitrification occurs in this region, so upwelled water is depleted with respect to nitrate. This benefits diazotrophs, who produce nitrate that is quickly recycled. Calcifiers have lower nutrient requirements than mixed phytoplankton, and disproportionately take advantage of the newly fixed nitrate. This

relationship does not establish in the NOCACO3TR model because in the absence of ballasting, high remineralization rates allow nutrients to be recycled closer to the surface, where denitrification does not occur.

    The mitigated response in CAL might also be partly driven by a substantially higher nitrate inventory (15% higher in RU and 32% in RD relative to MIXED; Table 2), which globally enhances NPP despite cooling temperatures (see the strongest and most widely distributed mixed phytoplankton RD response in CAL, Fig. A3). The causes of this enhanced inventory are

explored in Section 3.3.



## 3.2 Calcifiers and ballasting mitigate changes in suboxic volume and nitrate

A consequence of mitigated changes in carbon (and associated nutrient) export is the mitigated response of suboxic (defined here as below 5 mmol $O_2$) volumes (Fig. 3). In all models, ocean oxygen is consumed by POC remineralization, therefore smaller changes in POC produce smaller changes in oxygen consumption. In all RU models suboxic volume increases after

the transition to the more heterotrophic export regime, but in the CAL model the suboxic volume at year 500 is basically the same as at year 1, whereas in the other two models the suboxic volume has grown substantially (42% increase in MIXED, 75% increase in NOCACO3TR). Likewise in RD, in all models a decline in suboxic volume occurs with enhanced vertical mixing and ocean cooling, but with the CAL model experiencing the smallest decline (50% versus 56% in MIXED, 71% in NOCACO3TR by year 500).

Changes in suboxic volume regulate changes in denitrification (Fig. 3), which occurs in our model within suboxic zones. The mitigating effect of including a ballast model is apparent in a comparison of the denitrification rates in the three models RU and RD, where the smallest changes in denitrification occur in the CAL model (again, an eventual change less than one-third that of the NOCACO3TR response in both RU and RD). The smallest changes in nitrogen fixation also occur in the CAL model because the combination of ballasting and competition between calcifiers and diazotrophs mitigates increases in

the diazotroph population. Diazotrophs do not have as large a competitive advantage in CAL because the calcifiers efficiently export the phosphate out of their growth zones. However, a destabilizing effect on suboxia, denitrification and nitrogen fixation is apparent when including explicit calcifiers without ballasting (NOCACO3TR in Fig. 3). These larger changes in nitrate sources and sinks between icehouse and greenhouse biogeochemical regimes occur as a result of the competition between diazotrophs, mixed phytoplankton, and calcifiers. In RU the NOCACO3TR calcifiers expand in the low latitudes and produce

detritus, which remineralizes; this consumes oxygen in already low-oxygen regions, accelerating the expansion of suboxia and resultant denitrification. The calcifiers simultaneously benefit diazotroph populations by reducing the lateral export of phosphate out of the low latitudes, increasing nitrogen fixation by more than 6 Gmol N y$^{-1}$ by year 500. The largest responses in NOCACO3TR nitrate sources and sinks together produce the largest decline in ocean nitrate inventory in the NOCACO3TR model (about 900 Tmol N by year 500, not shown, compared to 800 Tmol N in MIXED and 100 Tmol N in CAL). In RD

the NOCACO3TR calcifiers rapidly decline, accelerating the contraction of suboxic volume and decline of denitrification. Nitrogen fixation also declines the most dramatically in NOCACO3TR (by more than 12 Gmol N y$^{-1}$ by year 500). However, the MIXED model experiences the largest increase in nitrate inventory in RD (a gain of 1800 Tmol N by year 500, not shown, compared to about 1650 Tmol N in NOCACO3TR and 500 Tmol N in CAL). This is due to the temporary increase in N fixation in the first 200 years of the MIXED simulation.

## 3.3 Calcifiers and ballasting enhance nitrate inventory and transfer of phosphate

In all models, changes in ocean circulation, as well as changes in biological rates, produce strong shifts in nutrient storage in both RU and RD oceans (Figs. 4 and 5). In RU, slowing circulation, a poleward shift in NPP, and increasing respiration rates move phosphate and nitrate into the deep ocean basins via the Southern Ocean (see discussion of the physical response





in Kvale et al. (2018)). The largest increases in deep ocean phosphate (zonal average increase of more than 0.6 mol m$^{-3}$) and nitrate concentrations occur in the Atlantic basin, which experiences a dramatic shoaling of overturning (from extending to the depth of the basin to extending less than 3000 m by year 150 Kvale et al., 2018). Antarctic Bottom Water (AABW) formation slows, which increases the residence time of particles and allows for more complete remineralization, increasing the dissolved

nutrient concentrations of the deep water masses. Phosphate is conserved in all models, therefore increased inventory in the deep oceans must also accompany decreased inventory in the upper ocean (up to about 1 mol m$^{-3}$ in zonal profiles). Strong declines in the upper ocean also occur in nitrate as suboxic volume increases, which increases denitrification rates (Fig. 3) faster than increasing diazotroph production can compensate. It is interesting that in spite of shoaling remineralization, which ought to increase the amount of nutrients held shallower in the water column, the net effect of climate warming is a deepening

of nutrient storage. Deeper nutrient storage contrasts the physics-dominated trend in DIC, which is an increase in DIC storage in the upper ocean (Kvale et al., 2018). The differences between DIC and phosphate storage trends illustrate the importance of where the changes in sources and sinks occur; the largest increases in biomass occur in the Southern Ocean (see Appendix Figure A2), where Antarctic Bottom Water (AABW) formation weakens (Kvale et al., 2018). An increased proportion of the global primary production occurring in the Southern Ocean, combined with a slowing of AABW, sequesters an increasing

proportion of the phosphate inventory in the deep ocean. However, ocean DIC uptake is more spatially distributed and thus storage has a different spatial pattern, at least under the transient conditions examined here. Model pre-existing DIC (DIC not due to rising atmospheric CO$_2$ concentrations) storage follows the spatial pattern of nutrients and is increasingly stored in the deep ocean, as described by DeVries et al. (2017).

    In RD, phosphate (Fig. 5) and nitrate concentrations rapidly increase (phosphate change of about 1 mol m$^{-3}$ in zonal

profiles) in the upper half of ocean basins and decrease (zonal average decrease of more than 0.6 mol m$^{-3}$) in the deeper regions. A deepening of ventilation and acceleration of meridional overturning (Kvale et al., 2018) mixes nutrient-rich deep water upwards and nutrient-depleted shallow water down in all models. The strongest response is again found in the Atlantic basin, where maximum overturning exceeds 40 Sv by year 150 (Kvale et al., 2018). In addition to accelerated overturning, a decline in NPP, paired with only small net increases in POC export production (and declines in PIC export) by year 500

in all models also encourages increased upper ocean nutrient storage in RD. Again, increased near-surface nutrient storage is counter-intuitive with a deepening remineralization length scale.

    In both RU and RD the CAL model displays the largest changes in phosphate and nitrate storage. This fact does not contradict the model mitigating changes in export fluxes. Maintenance of effective deep ocean export in CAL RU should logically increase deep ocean nutrient inventory. Maintenance of ineffective deep ocean export in CAL RD should logically increase shallow

ocean nutrient inventory. However, a separate process might also be driving the larger nutrient storage response in CAL. As was noted earlier, the CAL model contains a larger nitrate inventory than the other models (Table 2). This larger nitrate inventory is produced during model spin-up by a better spatial separation of nitrate production and denitrification. Close spatial association of these two processes has been demonstrated to lead to system-wide N loss (Landolfi et al., 2013). Competition between calcifiers, mixed phytoplankton, and diazotrophs increases the separation between these processes in NOCACO3TR,

leading to a modest increase in nitrate inventory. This increased separation occurs because calcifiers generally outcompete





mixed phytoplankton in the low latitude regions also inhabited by diazotrophs, but it is not a one-to-one replacement of mixed phytoplankton by calcifiers because calcifiers have lower nutrient requirements (this results in more calcifiers). These lower nutrient requirements cause calcifiers to also replace diazotrophs in regions where a faster growth rate produces an advantage (the regions between the gyres and the equatorial upwelling zones), which reduces the diazotroph population and separates

them spatially from denitrification regions. The CAL model additionally separates nitrogen fixation from denitrification in the vertical space, by reducing ocean suboxic volumes and lengthening the remineralization length scale via ballast protection. This reduction of "vicious N loss" (Landolfi et al., 2013) in the CAL model produces substantially higher nitrate inventory than in the other models in the greenhouse ocean. Because at an annual scale the models are almost universally nitrate-limited, this additional nitrate inventory enables a stronger NPP response in all phytoplankton types in RD CAL. More nitrate available

for photosynthesis means more phosphate is also taken up and therefore exported (or in the case of RD, rapidly remineralized).

## 4 Discussion

Taken together, RU and RD simulations using the three model configurations (CAL, NOCACO3TR, and MIXED) produce a lot of information on both icehouse and greenhouse climate transitions as well as biogeochemical controls related to the application of calcifiers and explicit PIC ballasting. These controls are summarized in Fig. 6. The icehouse ocean is characterized by low

atmospheric $CO_2$, deep overturning circulation dominated by the North Atlantic, a more autotrophic regime (lower rates of NPP with a high NPP:respiration ratio, distributed across latitudes, lower PIC and POC export; not shown, with a high proportion of POC exported to the deep ocean), low suboxic volume, high nitrate inventory, and shallower nutrient storage. The greenhouse ocean is characterized by high atmospheric $CO_2$, strong overturning circulation distributed between northern and southern hemispheres, a more heterotrophic regime (higher rates of NPP with a low NPP:respiration ratio, high production at polar

latitudes, high PIC and POC export; not shown, but a low proportion of the POC reaching the deep ocean), high suboxic volume, lower nitrate inventory, and deeper nutrient storage.

The biogeochemical transition between icehouse and greenhouse climate states follows a pathway that maps to changes in temperature and ocean circulation. All models' biogeochemistry becomes increasingly sensitive to the effects of temperature (changing biological rates) as they approach the greenhouse state, and increasingly sensitive to the effects of circulation

(changing nutrient availability) as they approach the icehouse state. The quasi-symmetric response of ocean phosphate and nitrate storage in moving between icehouse and greenhouse states presents a counter-intuitive example of competing temperature and circulation effects. In RU, remineralization rates increase and overturning circulation slows, which allows for more complete remineralization of particles. Therefore in RU, migration of NPP towards the poles (and the Southern Ocean particularly) produces a strong export of dissolved phosphate and nitrate to the deep ocean. Polar deep water formation regions have a high

sequestration efficiency (the length of time regenerated nutrients are stored in the interior before being returned to the surface, DeVries et al., 2012) and a high shallow ocean particle transfer efficiency (the fraction of POC export surviving respiration above 1000 m depth, DeVries and Weber, 2017), so increasing NPP in these regions increases the storage of nutrients in the deep water masses downstream. In RD, migration of NPP towards the low latitudes, as a response to the flushing of nutrients



from the deep ocean, produces a strong migration of phosphate to the upper middle latitude gyres and near-surface low latitudes. This occurs in spite of cooling temperatures reducing remineralization rates and a globally increased ocean ventilation, which together ought to reduce the amount of remineralization in the upper ocean and increase particle transfer efficiency. Even with declining remineralization rates, the particle transfer efficiency of the gyres and low latitudes remains low (the efficiency
is low even in a modern climate, DeVries and Weber, 2017), causing upper ocean nutrient concentrations to rise.

The spatial pattern in the shift in nutrient storage does not look like that of the dissolved inorganic carbon (DIC) anomaly from the same set of experiments, described in Kvale et al. (2018). DIC exchanges with the atmosphere and enters the ocean in RU through more distributed pathways, whereas phosphate and nitrate transport is strongly dominated by the Southern Ocean. However, RD anomalies of nutrient storage and DIC are more similar in that all share a common source of the deep
ocean. Similarly counter-intuitive anthropogenic carbon uptake responses to changing global circulation have been identified by DeVries et al. (2017). Theirs, and our, findings highlight the complexities of physical and biogeochemical changes that go beyond simple predictions of the global carbon and nutrient storage response to changing remineralization depths (e.g., Kwon et al., 2009; Meyer et al., 2016).

Inclusion of calcifiers and a ballast parameterization slows the rate of the transition in both directions. In RU, calcifiers out-
compete mixed phytoplankton in the warming low-latitudes in NOCACO3TR and CAL, maintaining NPP and PIC export while it declines in MIXED due to changes in circulation. As the model transitions towards heterotrophy, PIC ballasting mitigates rapid increases in NPP, PIC and POC export production, the expansion of suboxia and the loss of nitrate. However, explicit calcifiers without ballasting enhance the transition by increasing the NOCACO3TR model NPP and POC response to warming. In RD, explicit calcifiers without ballasting again enhance the transition by out-competing diazotrophs while heterotrophy still
dominates, rapidly decreasing NPP, POC and PIC production and export via surface nitrate limitation. However, inclusion of a ballast model in CAL slows the rate of transition in RD by reducing surface nitrate limitation. This mitigation of nitrate limitation occurs through two mechanisms: 1) strong separation of nitrogen fixation and denitrification regions maintains a higher global nitrate inventory, and 2) ballasting of nutrients by PIC maintains a lower nitrate surface concentration, which encourages the growth of diazotrophs. A secondary effect of the higher nitrate inventory in the CAL model is a stronger
response in phosphate storage to RU and RD transitions, as lower nitrate limitation increases the phytoplankton's ability to take up and export phosphate.

Unfortunately the idealized nature of our model simulations have no direct real-world analog and common understanding of greenhouse ocean nitrogen cycling is extremely limited (e.g., Kuypers et al., 2004; Junium and Arthur, 2007; Junium et al., 2018). Rapid warming of the Paleocene-Eocene Thermal Maximum (PETM; about 56 Ma) was accompanied by rapidly
expanding marine suboxia in the open ocean and some major reorganization of nitrogen cycling that left an isotopically similar signature to ocean anoxic events, possibly with an enhanced role of nitrogen fixers (Junium et al., 2018). Studies of $\delta^{15}$N for past ocean anoxic events suggest an increase in nitrogen fixation (Kuypers et al., 2004; Junium and Arthur, 2007), which would generally support our finding of overall higher nitrogen fixation in the greenhouse climate, and increasing N-fixation in a warming icehouse climate. The mechanisms we identify regarding nitrogen cycling broadly agree with those outlined by
Tyrrell (1999); Kuypers et al. (2004); Junium and Arthur (2007); enhanced denitrification rates produce an overall lower N





inventory, but high phosphate concentrations in upwelled water (our greenhouse ocean contains a vigorous overturning and deep ventilation) stimulate diazotrophy.

The dominance and distribution of coccolithophores have been shown to vary roughly proportionally to atmospheric $pCO_2$ over the past 50 million years, with coccolithophores being bigger, more dominant and widespread during periods of high
$pCO_2$ (Hannisdal et al., 2012). It has been hypothesized that efficient particle sinking by coccolithophores maintains their ideal growth conditions by increasing ocean oxygen, which increases the availability of metal cofactors required for their growth (Meyer et al., 2016). A similar feedback has also been speculated surrounding the evolutionary rise of eukaryotes; more efficient particle sinking efficiently oxygenated the global ocean, reduced phosphate inventory and further encouraged the development of complex life (Lenton et al., 2014). In our model we do not find total ocean oxygen to be higher in the CAL
model (not shown), but we do find the CAL model contains the smallest suboxic volume of the three model configurations. This suggests an additional stabilizing mechanism brought about by efficient particle export- the increased ocean inventory and stabilization of bioavailable nitrogen. Higher ocean inventory of nitrate potentially benefits most phytoplankton, but calcifiers in a greenhouse ocean would benefit disproportionately due to their lower nutrient requirements allowing them to thrive in oligotrophic conditions. Our model does not simulate changes in cell size, but higher calcification (resulting in larger particles
and even more efficient export) in a greenhouse climate could potentially magnify the benefit.

Our simulations present a highly idealized framework for exploring the multifaceted problem of ocean biogeochemistry and climate. As is discussed in Kvale et al. (2018), the RU simulation bears some resemblence to what might be expected in a rapid anthropogenically-induced climate warming scenario, but the RD simulation is more theoretical because any rapid decline in atmospheric $CO_2$ would be accompanied by a rapid flushing of ocean carbon to the atmosphere, thus maintaining high atmo-
spheric $CO_2$. Furthermore we do not critically analyze changes in ocean alkalinity, which might also exert a significant control on calcification over geological timescales (e.g., Bolton et al., 2016) (i.e. in greenhouse climates). Greenhouse climates likely also experienced different iron availability as well as weathering conditions, which we neglect in this study. Iron availability can affect the severity of the "vicious N cycle" (Moore and Doney, 2007) through the same spatial separation mechanism as additional competition by calcifiers. While we apply a static iron mask to both icehouse (RU) and greenhouse (RD) climates,
we do not simulate an evolving iron cycle.

## 5   Conclusions

Our model simulations demonstrate that calcifiers exert significant control on marine nitrogen cycling through a combination of competition and improved particle export efficiency. This control is apparent in both steady-state icehouse and greenhouse climates, in which calcifiers maintain a higher inventory of nitrate, as well as in transitions between states, in which the gain or
loss of calcifiers due to competition affects global particle export response to climate change. Application of calcifiers (including ballasting) slows the biogeochemical transition between more autotrophic icehouse and more heterotrophic greenhouse productivity regimes because of this control on the nitrogen cycle.




We additionally demonstrate the importance of considering both changes in biological rates as well as changes in physical circulation state in the simulation of net biogeochemical response to climate forcing. Accumulation of nutrients in the deep ocean in a greenhouse climate, in spite of shoaling remineralization profiles, and accumulation of nutrients in the upper ocean in an icehouse climate, in spite of lengthening remineralization profiles, are surprising results. The importance of changing

biological rates, as well as physical environment, to transitions between more autotrophic and more heterotrophic productivity regimes is also apparent in the quasi-symmetry of the global NPP transition we describe.

*Code and data availability.*  Model data and postprocessing scripts are available from https://thredds.geomar.de/thredds/catalog/open_access/ kvale_etal_2018_bg/catalog.html. Model code is available from the authors upon request.

## Appendix A:  Transitionary response of global integrated NPP

Global integrated net primary production (NPP) changes little with increasing sea surface temperatures (SSTs) in RU (Fig. A1) until after global average SSTs exceed 22°C, or after about 3.5°C change. We note the similarity in the RU/RD transition point SST anomaly but cannot identify it as a tipping point without additional tests of different rates of forcing and equilibrium states. The drivers of NPP response before and after this threshold temperature change are explored in Kvale et al. (2015b). In Kvale et al. (2015b) the temperature threshold was found to be between 2 and 4°C average SST change; the clearer transition

point in the present study is likely due to the more structured nature of the $CO_2$ forcing used here (rapid increase then a sudden stabilization). Before the transition point, the ocean circulation response to increasing atmospheric $CO_2$ drives the NPP trend (Kvale et al., 2015b). Surface stratification limits nutrient resupply from the deep ocean, and nutrient limitation spreads. The MIXED model shows a slight decline in NPP with increasing SSTs because the mixed phytoplankton are increasingly phosphate limited in the low latitudes (Fig. A2). The models including explicit calcifiers do not experience a global decline

in NPP because the calcifiers are able to take advantage of spreading nutrient limitation and their range expands (Fig. A2 and Kvale et al. 2015b). After the threshold 3.5°C SST change is crossed in RU, enhanced upper-ocean biological rates become an increasingly important driver of global NPP (Kvale et al., 2015b). NPP rapidly increases in all model configurations (Fig. A1). Likewise, the ratio of NPP to upper-ocean (top 130 meters) heterotrophic rates (a total of microbial fast recycling, zooplankton digestion, and particle remineralization, termed 'respiration' here and in Kvale et al. 2015b), declines at an accelerating rate

(Fig. A1). Rapid remineralization shoals the remineralization length scale, quickly recycling nutrients higher in the water column and making them available to primary producers on a shorter time scale. Mixed phytoplankton in the Southern Ocean become increasingly nutrient, rather than light, limited (not shown). In CAL and NOCACO3TR, calcifiers expand their range in the north Atlantic. In all models mixed phytoplankton increase in concentration in both polar oceans and in the southern hemisphere middle latitudes. Diazotrophs increase in the low latitudes. Zooplankton respond to increasing food availability

and their biomass increases in the high latitudes and southern hemisphere.





In RD, global NPP demonstrates a more model-specific response, though a threshold temperature is again apparent below 21°C, or after about 3.5°C of global average SST cooling (read the green lines right-to-left in Fig. A1). As in RU, the early period of relatively small changes in global NPP lasts roughly until $CO_2$ forcing is stabilized, and is again driven by a combination of physical and biological processes. Slight increases in global NPP occur in the CAL and MIXED models as the global

trend is dominated by an increase in low and middle-latitude northern hemisphere mixed phytoplankton biomass (Fig. A3). NPP increases in this region, as a response to the upward mixing of nutrients that occurs with the acceleration of meridional overturning, combined with relatively high SSTs (Kvale et al., 2018) that maintain high rates of respiration (Fig.A1). A decline in global NPP in the first 150 years of RD in the NOCACO3TR model reflects the decline in diazotrophs in the low latitudes over this time (Fig. A3). All models experience rapid declines in global NPP, and rapid increases in the NPP:respiration ratio,

after the SST threshold is crossed, as cooling temperatures and slowing overturning (Kvale et al., 2018) reduce nutrient availability and biological rates. We interpret this to indicate a transition into a more "autotrophic" regime, in which near-surface respiration has a reduced role in driving primary production.

In the MIXED model, diazotrophs respond strongly to increasing surface phosphate concentrations. The nitrate inventory is smaller in this model than in the other two (see Table 2 and described below), which places mixed phytoplankton at more of

a relative disadvantage with respect to diazotrophs than in the other models. However, mixed phytoplankton exploit the nitrate the diazotrophs add to the near-surface so effectively in the warm seawater that in parts of the low latitudes they are phosphate-limited at the start of RD (not shown). This phosphate limitation subsides as more deep phosphate reaches the surface ocean, and as sea surface temperatures cool and NPP slows.

In NOCACO3TR, inclusion of calcifiers increases ecological competition between phytoplankton types. Calcifiers require

both nitrate and phosphate for production, which is a handicap with respect to diazotrophs, but their growth rate is faster. Calcifiers grow slower relative to mixed phytoplankton, but take advantage of lower nutrient requirements. The net response in calcifier biomass is a small increase in the first 150 years of RD in the northern hemisphere and along the Great Calcite Belt (an observed region of high coccolithophore population density around 40°S; Balch et al. 2016), and small decreases everywhere else. An increase in calcifiers mitigates the positive diazotroph response to phosphate fertilization from enhanced vertical

mixing because the calcifiers consume the phosphate first. Diazotrophs suffer a net decline in biomass. Mixed phytoplankton nitrate limitation is therefore more ubiquitous than in the MIXED model. Calcifiers are replaced by mixed phytoplankton in the Arctic. The net change in NPP is a decline that accelerates after the SST threshold is reached.

Figure A1 clearly reveals the role of ballast by PIC in determining total NPP. In both RU and RD, the total NPP is lower when the ballast model is applied than in the other model configurations. This is because ballasting lowers total nutrient

concentrations near the surface by effectively exporting particles to the deep sea for remineralization, thereby limiting all primary producers. The global integrated NPP response of the CAL model in RD is lower in magnitude but similar in slope, to that of MIXED. Like in MIXED, diazotroph biomass increases in the low latitudes and produces nitrate that is readily available for other phytoplankton to use. Like NOCACO3TR, calcifier biomass increases along the Great Calcite Belt and in the northern hemisphere, and declines in the middle latitudes and around the Equator. Mixed phytoplankton biomass increases

almost globally and shows the largest biomass increases in any of the three models. However the net NPP response is a decline





after the threshold 3.5°C SST change, which suggests the increase in mixed phytoplankton is not offset by the large decrease in calcifiers, and the reduction in primary production rates. In CAL, diazotroph biomass does not experience the large decline seen in the other two models. This is due to the effect of ballasting, which maintains a lower surface nitrate concentration and thus a more favorable environment for diazotrophs. In all models, an equatorward shift in zooplankton biomass occurs over the

5   RD as the predators follow their prey, opposite to the poleward shift seen in RU.

*Author contributions.*   KFK designed the experiment, and wrote the paper with comments by the other authors. KET ran the simulations and contributed to the analysis of the data. AL and KJM contributed to the analysis.

*Competing interests.*   The authors declare no competing interests.

**Data availability**

10   All model output used to produce this manuscript is available at: https://thredds.geomar.de/thredds/catalog/open_access/kvale_ et_al_2018_bg/catalog.html. Model code is available from the corresponding author upon request.

*Acknowledgements.*   This work was supported by GEOMAR Helmholtz Centre for Ocean Research Kiel and Sonderforschungsbereich 754. KT and KFK are grateful for computing resources provided by GEOMAR and Kiel University. KJM acknowledges support from the Australia Research Council (DP180100048).



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





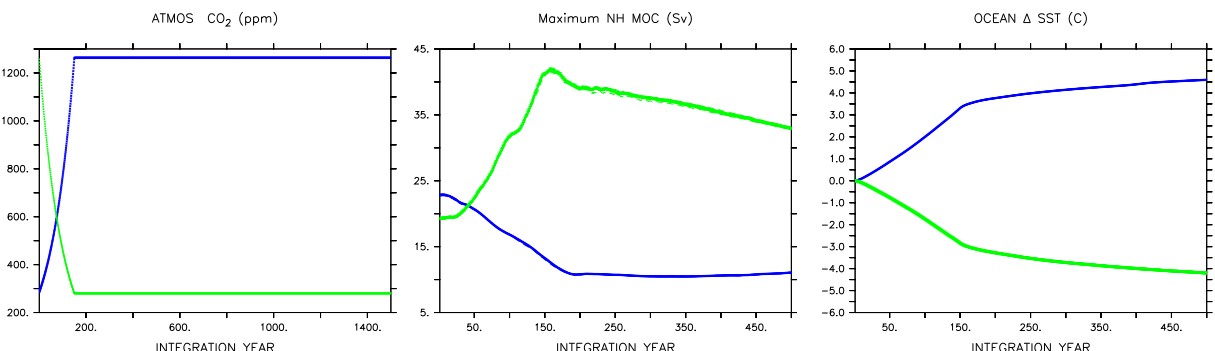

**Figure 1.** Atmospheric $CO_2$ forcing used for RU icehouse (blue) and RD greenhouse (green) experiments (left panel). Maximum northern hemisphere meridional overturning (middle plot), and change in ocean average sea surface temperature (right plot). Dotted lines represent the MIXED model, thin dashed lines represent the NOCACO3TR model, and thick dashed lines represent the CAL model. The physical responses of the models to RU and RD forcing are nearly identical.





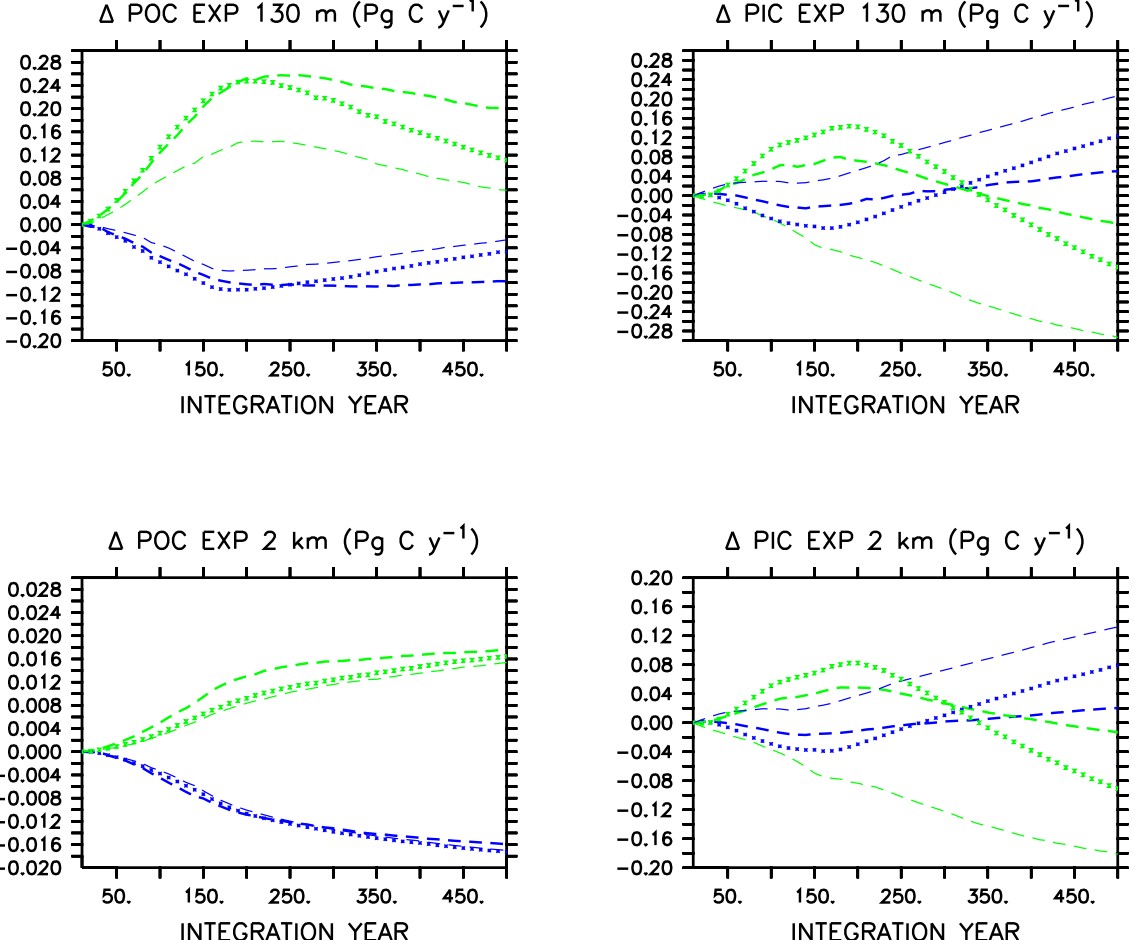

**Figure 2.** Change in globally integrated annual POC (left column) and PIC (right column) export in RU icehouse (blue) and RD greenhouse (green) experiments. Dotted lines represent the MIXED model, thin dashed lines represent the NOCACO3TR model, and thick dashed lines represent the CAL model.



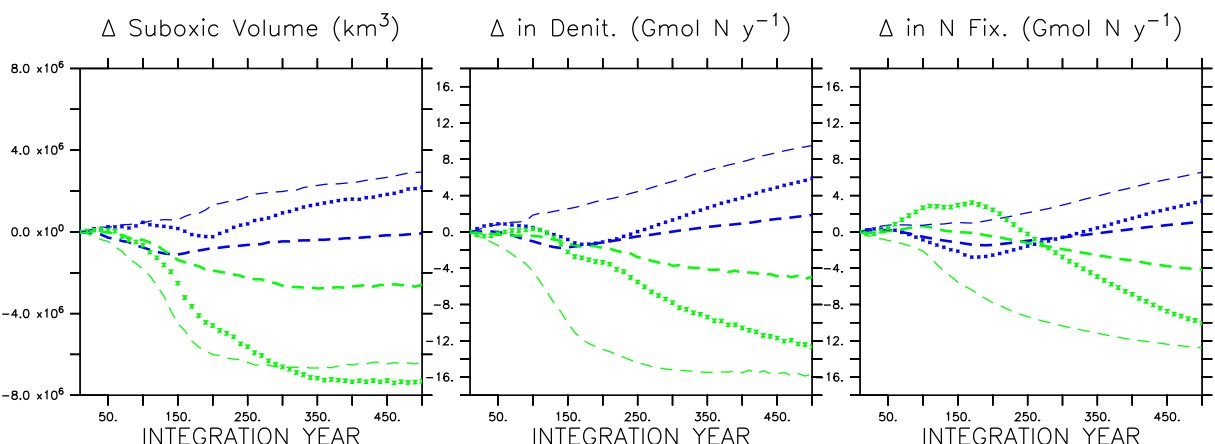

**Figure 3.** Change in globally integrated suboxic volume in RU icehouse (blue) and RD greenhouse (green) experiments (left). Change in globally integrated denitrification (middle). Change in globally integrated nitrogen fixation (right). Dotted lines represent the MIXED model, thin dashed lines represent the NOCACO3TR model, and thick dashed lines represent the CAL model.





**Figure 4.** Zonally averaged changes in phosphate concentration over the RU simulation. Bottom row shows changes at surface.





**Figure 5.** Zonally averaged changes in phosphate concentration over the RD simulation. Bottom row shows changes at surface.



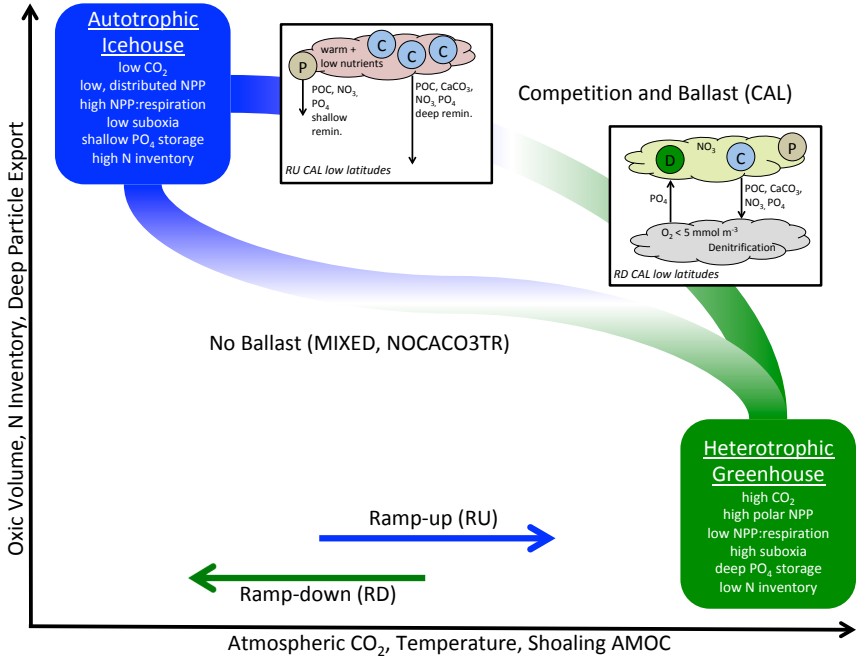

**Figure 6.** A qualitative depiction of biogeochemical controls of calcifiers in RU and RD simulations. The biogeochemical transition between the two productivity regimes (y axis) occurs along a pathway of changes to temperature and ocean circulation, which are driven exclusively by forcing of atmospheric $CO_2$ in our simulations (x axis). Calcifiers control the pace of the biogeochemical transition through two mechanisms: PIC ballasting of POC and ecological competition. Application of explicit calcifiers and a PIC ballast model slows the pace of transition in both directions, as reduces the magnitude of the total equilibrated change in suboxic volume, N inventory, and particle export. Application of explicit calcifiers, without ballasting, enhances the pace of transition along the pathway in both directions. Inset into the figure are diagrams depicting the low latitude nutrient exchange between calcifiers (C) and mixed phytoplankton (P) in RU, and diazotrophs (D) and calcifiers (C) in the CAL model in RD. Calcifiers export carbon and nutrients with both ballasted and non-ballasted particles. In RU, calcifiers benefit from warming and stratification and efficiently export particles for deep ocean remineralization and dissolution. This inhibits the recycling of nutrients in the near-surface to form regenerated production by other phytoplankton (P), slowing the transition into greenhouse climate conditions. In RD, low latitude denitrification reduces the nitrate content in upwelled water, which promotes the growth of diazotrophs. The fixed nitrate is used by the calcifiers and thus both phytoplankton types benefit each other. Mixed phytoplankton (P) also benefit from nitrate fertilization, though the effect is smaller.





**Table 1.** Summary of ecosystem components for each model configuration. Abbreviations stand for: particulate organic carbon (POC), calcium carbonate ($CaCO_3$), mixed phytoplankton (P), diazotrophs (D), zooplankton (Z), small phytoplankton and calcifiers (C).

| Model | Carbon Export Tracers | Plankton Functional Types |
|---|---|---|
| MIXED | Free POC | P, D, Z |
| NOCACO3TR | Free POC | P, D, C, Z |
| CAL | Free POC, Ballast POC, $CaCO_3$ | P, D, C, Z |

**Table 2.** Initial ocean nitrate inventory in Pmol N.

| Model | 285 ppm $CO_2$ | 1257 ppm $CO_2$ |
|---|---|---|
| MIXED | 37.11 | 29.79 |
| NOCACO3TR | 39.45 | 32.23 |
| CAL | 42.66 | 39.10 |

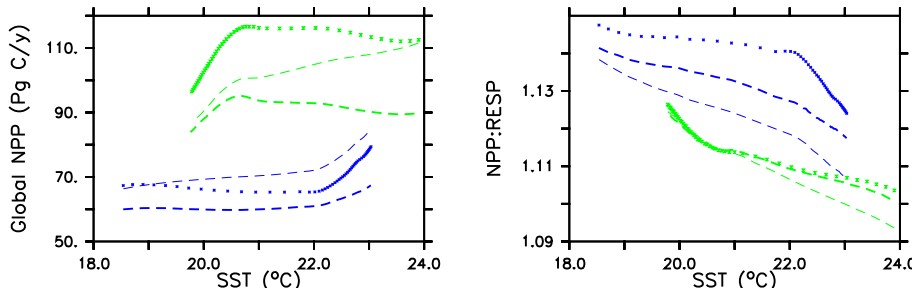

**Figure A1.** Globally integrated annual net primary production and sea surface temperature in RU icehouse (blue) and RD greenhouse (green) experiments, left figure. Globally integrated annual NPP divided by gross respiration and sea surface temperature, right figure. Dotted lines represent the MIXED model, thin dashed lines represent the NOCACO3TR model, and thick dashed lines represent the CAL model.





**Figure A2.** Zonally averaged changes in plankton biomass over the RU simulation.







**Figure A3.** Zonally averaged changes in plankton biomass over the RD simulation.