# Peer review of "Phytoplankton calcifiers control nitrate cycling and the pace of transition in warming icehouse and cooling greenhouse climates"

_Biogeosciences, 2018_

## Referee Comment (RC1) · Anonymous Referee #1 · 3 Dec 2018

This paper describes simulation experiments with the EMIC UVic v2.9, in which atmospheric $CO_2$ is prescribed in a warming (ramp-up, or RU) and a cooling (ramp-down, or RD) scenario, during which $CO_2$ changed by 1% per year for 150 years. Three different model versions with different details of the biological carbon pump (in detail, the implementation of coccolithophores) are compared and results of the marine carbon cycle are discussed — with respect to the differences of the warming and cooling and also with respect to the importance of the details in the biological pump.

This paper is a follow-up of a paper just published in ERL, in which the same warming and cooling experiments have been analysed, but with only one set-up of the biological

carbon pump. The results are interesting and I found no major resason against its publication. However, there is a list of minor issues which I like the authors to go through in order to come to an improved version of the manuscript.

- Increasing atm $CO_2$ from 285 ppm by 1%/y is similar to $CO_2(t) = CO_2(0) \times (1.01)^t$, which gives for $t = 150$ years a values of 1268 ppm, while the authors end their ramp-up experiments at 1257 ppm. Similarly, ramping down from 1257 ppm for 150 years gives after $CO_2(t) = CO_2(0) \times (0.99)^t$ a $CO_2$ of 278 ppm, not 285 ppm which the authors get. So, something in the described $CO_2$ scenario is corrupted. Since one paper with the same forcing has already been published, I suggest it is enough to refine the descripion of the forcing, I do not think new experiments are necessary. Maybe this misfit can be easily solved (e.g. rounding error?), if so, explain it.

- The scenarios are called RD (ramp-down) and RU (ramp-up) here, but have been called COOLING and WARMING in the initial paper (Kvale et al 2018, ERL). I suggest that the authors stick to the original names, which would then make it much easier for the readers to follow both papers.

- The description of the 3 different model configurations is too short and weak. What is the difference between the scenarios CAL and NOCACO3TR? In the text it is written, that NOCACO3TR does not contain prognostic CaCO3 tracer. What does this imply? For my understanding, the tracer might be only an output variable, but it seems that it is also that parts of the model are different. How is this related to ballasting (which is given as motivation for this study in the introduction)? I believe the word "ballasting" is not mentioned in the methods section at all. One gets some ideas of what is different from Table 1, but this should be expanded in the text of the methods section.

- page 2, line 6. ... PIC production of 1-1.6 PgC/y should be only 1-3% of total annual POC export. This would imply that annual POC export is around 100

PgC/yr. This is a number which I believe is much too high. However, to really evaluate it one needs to know how the authors define export production — which the reader does not yet know. Typically this is the vertical flux around a water depth of 100 m, but sometimes other depths are taken and then fluxes have to be transformed (using assumptions on remineralisation rates) to make them comparable. POC export at 100 m water depth is around 10 PgC/y (e.g. see Table 4 in Laufkötter et al., 2016), not 100 PgC/y. Please revise, explain and probably correct. Please state once, to which water depth all calculations of export prodcutions refer to.

- page 5, line 7: The ramp-up experiment leads to a warming of 6.8 K. Since in the ramp-up $CO_2$ rises from 285 ppm to 1257 ppm a short notation on the climate sensitvitiy (equilibrium temperature rise for $2xCO_2$) of the model would be helpful to set this into context with other models. It is furthermore said, that zonally averaged upper ocean warms by as much as 8.6 K. This would imply the ocean warms more than the atmosphere, which is difficult to understand, when the $CO_2$ rise is the initial driver for the temperature rise. Is this connected with ocean circulation changes? If so, are there areas in which the ocean cools? Maybe the average ocean warming might also help here. Please explain.

- page 5, line 28-30: *However, the inclusion of PIC ballast in CAL causes near-surface POC export not to increase because the expanding calcifier population provides protection to an increasing proportion of the total POC from near-surface remineralization (Fig. 2 and Kvale et al., 2015b).* This is opposite of what I would think. If near-surface POC is protected from remineralisation, then POC export should increase. Please check and explain.

- page 6, lines 18-19. Define shallow and deep POC export.

- page 7, line 2: "5 mmol O2" misses some units, probably 5 mmol $O2/m^3$.

- Appendix A (Transitionary response of global integrated NPP): This should become part of the main text, e.g. start the result section with it, or if only of minor relevance be deleted.

- Reference list: Most papers have long and double entries with links to the papers. This should be reduced to one entry with the DOI, or at max with a full www address including the DOI (https://doi.org/...). Please check all links, some are corrupted, including the link to the previous paper (Kvale et al 2018 in ERL).

- Ref to Balch et al 2018 is a journal paper, but contains editors, which is weird for journal entries, please correct.

- Fig 1: Fonts of x- and y-axes labels are too small, maybe order subfigures vertically, not horrizontally, and therefore be able to increase size of the figures.

- Fig 1c. These are only changes in temperature, please give also absolute values somewhere.

- Table 2: Why are the initial nitrate inventory different for the 6 runs? Is there a tuning target, which was achieved, if so, give details in methods. And what does these differences in the nitrate inventory imply for the comparision of the runs?

**References**

Laufkötter, C., Vogt, M., Gruber, N., Aumont, O., Bopp, L., Doney, S. C., Dunne, J. P., Hauck, J., John, J. G., Lima, I. D., Seferian, R., and Völker, C.: Projected decreases in future marine export production: the role of the carbon flux through the upper ocean ecosystem, Biogeosciences, 13, 4023–4047, doi:10.5194/bg-13-4023-2016, 2016.

---

## Referee Comment (RC2) · Anonymous Referee #2 · 16 Jan 2019

Review of manuscript: "Phytoplankton calcifieres control nitrate cycling and the pace of transition in warming icehouse and cooling greenhouse climates" by Karin F. Kvale, Katherine E. Turner, Angela Landolfi, and Kathrin J. Meissner

In their manuscript the authors address the question which role phytoplankton calcifieres might play during rapid climate transitions on centennial time scales in affecting oceanic tracer distributions and the marine biogeochemical cycles. Therefore they have employed an Earth system model of intermediate complexity ( UVic ) which was recently upgraded by implementing coccolithophorides and biogenic calcite as a fully prognostic tracer, which also accounts for the mineral ballast effect. By ramping atmospheric pCO2 levels up (RU) and down (RD) between 285 and 1257 ppm in their experiments, the model was driven from icehouse to greenhouse conditions and vice versa. The main result of the study is the discovery of a mitigation effect of PIC ballast on several biogeochemical tracers. As an example, in the RU scenario which runs the model from a low CO2 icehouse to high CO2 greenhouse conditions, an increasing ballast effect due to enhanced PIC production mitigates changes in export fluxes and prevents the development of augmented oxygen minimum zones (OMZ). The paper is interesting, well organized and clearly written. Therefore, subject to minor revisions, I can recommend publication in Biogeosciences.

General comment: As mentioned in the manuscript, and in contrast to earlier studies, recent literature points towards elevated coccolithophoride production rates under rising temperatures and ocean acidification. However, this is not necessarily the case for biogenic calcification. Keeping temperature, alkalinity and nutrient concentrations fixed, increasing CO2 concentrations will lead to diminished calcification rates in many of the coccolithophoride species (see Bach et al., 2015). As a result, reduced calcification rates under high CO2 conditions could counteract the overall stimulating effects on coccolithophorides and weaken the PIC ballasting effect, which then could even lead to a spread of OMZs, as found by Hofmann and Schellnhuber (2009). To my knowledge, UVic does not account for a pH-value (or Omega) dependent calcification rate. Therefore, I would appreciate a short paragraph in the manuscript clarifying this issue.

Minor comments: page 2 line 3 : The sentence "... marine carbon export production is well-established Balch (recently summarized by 2018)" should be rewritten as : ".. marine carbon export production is well-established (recently summarized by Balch, 2018)"

page 7 lines 2 and 3: " ... suboxic (defined here as below 5 mmol O2)"; did you mean 5 mmol/m3 O2 ?

pages 18 and 20: The labels on axis of figures 1 and 3 are rather small, please enlarge.

Page 12 lines 18-21: In the sentence starting with: "The MIXED model shows a slight decline in NPP ..." refers to figure A2, which shows plankton biomass but no NPP. Please correct this mismatch or rephrase the sentence.

Reference:

Bach, L.T., et al. A unifying concept of coccolithophore sensitivity to changing carbonate chemistry embedded in an eco- logical framework. Prog. Oceanogr. (2015), http://dx.doi.org/10.1016/j.pocean.2015.04.012

---

## Author Comment (AC1) · 6 Feb 2019

Phytoplankton calcifiers control nitrate cycling and the pace of transition in warming icehouse and cooling greenhouse climates

Author response to RC1

The authors would like to thank the Referee for their careful reading of our manuscript and thoughtful comments. Our responses are given below. Major modifications to the manuscript text are reproduced in red font.

**This paper describes simulation experiments with the EMIC UVic v2.9, in which atmospheric CO2 is prescribed in a warming (ramp-up, or RU) and a cooling (ramp-down, or RD) scenario, during which CO2 changed by 1% per year for 150 years. Three different model versions with different details of the biological carbon pump (in detail, the implementation of coccolithophores) are compared and results of the marine carbon cycle are discussed — with respect to the differences of the warming and cooling and also with respect to the importance of the details in the biological pump. This paper is a follow-up of a paper just published in ERL, in which the same warming and cooling experiments have been analysed, but with only one set-up of the biological carbon pump. The results are interesting and I found no major reason against its publication. However, there is a list of minor issues which I like the authors to go through in order to come to an improved version of the manuscript.**

**• Increasing atm CO2 from 285 ppm by 1%/y is similar to CO2(t) = CO2(0)×(1.01)t, which gives for t = 150 years a values of 1268 ppm, while the authors end their ramp-up experiments at 1257 ppm. Similarly, ramping down from 1257 ppm for 150 years gives after CO2(t) = CO2(0)×(0.99)t a CO2 of 278 ppm, not 285 ppm which the authors get. So, something in the described CO2 scenario is corrupted. Since one paper with the same forcing has already been published, I suggest it is enough to refine the description of the forcing, I do not think new experiments are necessary. Maybe this misfit can be easily solved (e.g. rounding error?), if so, explain it.**

Examination of the equilibrated output files shows the models were initiated at 283.9 and 1263 ppm.

This Matlab script produces a value of 1250 ppm at year 150, and 1263 ppm at year 151 (rounded to the nearest whole value):
```
co2=283.9;
for i=2:151
co2(i) = co2(i-1)+co2(i-1)*0.01;
end
```

And this one produces a value of 283 ppm at year 150, and 280 ppm at year 151 (rounded to the nearest whole value):
```
co2 = 1263;
for i=2:151
```

```
co2(i) = co2(i-1)+co2(i-1)*0.01;
end
```

Examination of the CO2 in the data files agrees with the Matlab script, with some rounding difference. The discrepancy in the paper can be explained in that year 1 in the ramp-up data files is 285 ppm, and year 1 in the ramp-down data files is 1257 ppm (283.9 and 1263 ppm is year 0). So while the change in forcing was applied in a symmetric way from year 0, the end points do not match in the case of the ramp-down (280 versus 283.9 ppm) and the actual stabilisation point was after 151 years.

The CO2 concentration values and simulation forcing time values in the manuscript text have been corrected.

**• The scenarios are called RD (ramp-down) and RU (ramp-up) here, but have been called COOLING and WARMING in the initial paper (Kvale et al 2018, ERL). I suggest that the authors stick to the original names, which would then make it much easier for the readers to follow both papers.**

The naming has been made consistent with the other paper.

**• The description of the 3 different model configurations is too short and weak. What is the difference between the scenarios CAL and NOCACO3TR? In the text it is written, that NOCACO3TR does not contain prognostic CaCO3 tracer. What does this imply? For my understanding, the tracer might be only an output variable, but it seems that it is also that parts of the model are different. How is this related to ballasting (which is given as motivation for this study in the introduction)? I believe the word "ballasting" is not mentioned in the methods section at all. One gets some ideas of what is different from Table 1, but this should be expanded in the text of the methods section.**

The CAL model prognostic calcite and ballasting parameterisation is described in the Introduction, from page 2 line 23. Calcite influences model outcome in the CAL model through the export of POC and PIC. New sentences are added to the Methods section, page 4 line 17:

This prognostic CaCO3 tracer includes a POC ballasting parameterisation in which a fixed fraction of calcifier-associated POC is "protected" from water column remineralisation and exported from the surface at the CaCO3 sinking rate. It is released back into the "free" POC pool at the rate of dissolution of CaCO3.

**• page 2, line 6. … PIC production of 1-1.6 PgC/y should be only 1-3% of total annual POC export. This would imply that annual POC export is around 100 PgC/yr. This is a number which I believe is much too high. However, to really evaluate it one needs to know how the authors define export production — which the reader does not yet know. Typically this is the vertical flux around a water depth of 100 m, but sometimes other depths are taken and then fluxes have to be transformed (using**

**assumptions on remineralisation rates) to make them comparable. POC export at 100 m water depth is around 10 PgC/y (e.g. see Table 4 in Laufkötter et al., 2016), not 100 PgC/y. Please revise, explain and probably correct. Please state once, to which water depth all calculations of export prodcutions refer to.**

The paragraph is edited to distinguish between export and production (page 2, lines 3-6).

**• page 5, line 7: The ramp-up experiment leads to a warming of 6.8 K. Since in the ramp-up CO2 rises from 285 ppm to 1257 ppm a short notation on the climate sensitivity (equilibrium temperature rise for 2xCO2) of the model would be helpful to set this into context with other models. It is furthermore said, that zonally averaged upper ocean warms by as much as 8.6 K. This would imply the ocean warms more than the atmosphere, which is difficult to understand, when the CO2 rise is the initial driver for the temperature rise. Is this connected with ocean circulation changes? If so, are there areas in which the ocean cools? Maybe the average ocean warming might also help here. Please explain.**

For these simulations we have not equilibrated the models at a doubling of CO2 in order to diagnose an equilibrium climate sensitivity. We can therefore not make any statement on equilibrium climate sensitivity, because the models are far from equilibrium.

The sentence in question gives the global average SAT rise and a zonal average upper ocean temperature change. The global del SAT is provided for context, but a zonal average ocean temperature is given because it is more relevant for biology. The sentence is changed to emphasize the difference (page 5 line 8): Global average surface air temperature increases by more than 6.8°C, but the maximum of zonal average upper ocean temperatures increases by as much as 8.6°C by year 500 (Kvale et al., 2018).

**• page 5, line 28-30: However, the inclusion of PIC ballast in CAL causes near- surface POC export not to increase because the expanding calcifier population provides protection to an increasing proportion of the total POC from near-surface remineralization (Fig. 2 and Kvale et al., 2015b). This is opposite of what I would think. If near-surface POC is protected from remineralisation, then POC export should increase. Please check and explain.**

The reviewer is correct in the case of deep ocean export, but in the near-surface the result is a little less intuitive. In the CAL model, POC producers include calcifiers, mixed (non-calcifying) phytoplankton, and diazotrophs. Surface warming and stratification favours calcifiers. In the CAL model, POC produced by calcifiers is exported from the surface with a much longer remineralisation length scale as ballast. All other POC is exposed to remineralisation as soon as it forms. Kvale et al 2015b describes in detail how an increase in "free" POC in the near-surface quickly remineralises in warming conditions, fueling additional

"free" POC production and near-surface export. Deep ocean "free" POC export decreases because the POC remineralises high in the water column. By including ballast POC, when calcifiers are favoured they take up a larger proportion of the total biomass, which means an increasing amount of the total POC is in the "ballast", rather than the "free" pool. This ballast POC is able to reach the deep ocean, where it does not drive a large increase in NPP (and hence, POC production) at the surface.

This process is described starting from line 30 in the manuscript, and in greater detail in Kvale et al 2015b.

• **page 6, lines 18-19. Define shallow and deep POC export.**

Done.

• **page 7, line 2: "5 mmol O2" misses some units, probably 5 mmol O2/m3.**

Fixed.

• **Appendix A (Transitionary response of global integrated NPP): This should be- come part of the main text, e.g. start the result section with it, or if only of minor relevance be deleted.**

The authors believe this section is interesting for a sub-section of readers and adds context to the paper, but is not part of the main point (calcifiers mitigating biogeochemical changes). Earlier drafts included it as part of the main body, but we found this dilutes and confuses the main message of the paper. We prefer to leave it as an appendix because it is not enough to stand as a paper of its own and is relevant for understanding the main analysis. It would also be useful for our ongoing work on this subject to have the appendix published along with the manuscript.

• **Reference list: Most papers have long and double entries with links to the papers. This should be reduced to one entry with the DOI, or at max with a full www address including the DOI (https://doi.org/...). Please check all links, some are corrupted, including the link to the previous paper (Kvale et al 2018 in ERL).**

Fixed.

• **Ref to Balch et al 2018 is a journal paper, but contains editors, which is weird for journal entries, please correct.**

Fixed.

• **Fig 1: Fonts of x- and y-axes labels are too small, maybe order subfigures vertically, not horizontally, and therefore be able to increase size of the figures.**

**• Fig 1c. These are only changes in temperature, please give also absolute values somewhere.**
This figure is removed to make more space for the other two. It was not needed as it was never discussed in the manuscript.
**• Table 2: Why are the initial nitrate inventory different for the 6 runs? Is there a tuning target, which was achieved, if so, give details in methods. And what does these differences in the nitrate inventory imply for the comparison of the runs?**

 Initial nitrate inventories are different for each model and model set-up because this quantity is not conserved. The models were not tuned with respect to nitrate. Section 3.3 details the implications of different nitrate inventories. A sentence is added to the Methods section (page 4 line 30):
Ocean nitrate concentration is allowed to evolve from the balance between nitrogen source (phytoplankton nitrogen fixation) and sink (denitrification).

---

## Author Comment (AC2) · 6 Feb 2019

Phytoplankton calcifiers control nitrate cycling and the pace of transition in warming icehouse and cooling greenhouse climates

Author response to RC2

The authors would like to thank the Referee for their careful reading of our manuscript and thoughtful comments. Our responses are given below. Major modifications to the manuscript text are reproduced in red font.

**Review of manuscript: "Phytoplankton calcifieres control nitrate cycling and the pace of transition in warming icehouse and cooling greenhouse climates" by Karin F. Kvale, Katherine E. Turner, Angela Landolfi, and Kathrin J. Meissner**
**In their manuscript the authors address the question which role phytoplankton calcifieres might play during rapid climate transitions on centennial time scales in affecting oceanic tracer distributions and the marine biogeochemical cycles. Therefore they have employed an Earth system model of intermediate complexity ( UVic ) which was recently upgraded by implementing coccolithophorides and biogenic calcite as a fully prognostic tracer, which also accounts for the mineral ballast effect. By ramping atmospheric pCO2 levels up (RU) and down (RD) between 285 and 1257 ppm in their experiments, the model was driven from icehouse to greenhouse conditions and vice versa. The main result of the study is the discovery of a mitigation effect of PIC ballast on several biogeochemical tracers. As an example, in the RU scenario which runs the model from a low CO2 icehouse to high CO2 greenhouse conditions, an increasing ballast effect due to enhanced PIC production mitigates changes in export fluxes and prevents the development of augmented oxygen minimum zones (OMZ). The paper is interesting, well organized and clearly written. Therefore, subject to minor revisions, I can recommend publication in Biogeosciences.**

**General comment: As mentioned in the manuscript, and in contrast to earlier studies, recent literature points towards elevated coccolithophoride production rates under rising temperatures and ocean acidification. However, this is not necessarily the case for biogenic calcification. Keeping temperature, alkalinity and nutrient concentrations fixed, increasing CO2 concentrations will lead to diminished calcification rates in many of the coccolithophoride species (see Bach et al., 2015). As a result, reduced calcification rates under high CO2 conditions could counteract the overall stimulating effects on coccolithophorides and weaken the PIC ballasting effect, which then could even lead to a spread of OMZs, as found by Hofmann and Schellnhuber (2009). To my knowledge, UVic does not account for a pH-value (or Omega) dependent calcification rate. Therefore, I would appreciate a short paragraph in the manuscript clarifying this issue.**

This is an important point that was missing from the Discussion section. The following has been added (page 11, line 10):

It is important to remind the reader that the UVic ESCM does not currently account for any potential effect of pH or alkalinity on biological calcification. Therefore increases (decreases) in primary production result in corresponding increases (decreases) in calcium carbonate production. This assumption might or might not be valid at a global scale, in which a large diversity of calcifier species are exposed to rapid changes in pH and alkalinity (e.g., Balch 2018, Krumhardt et al 2017, Monteiro et al 2016). How rates of forcing compare to rates of changes to global carbonate weathering might also determine the legitimacy of our assumption (Bach et al 2015). We acknowledge that pH-dependent calcification could lead to different sensitivities that might result in different conclusions than what we find here (i.e., the CAL model would respond to forcing more like the NOCACO3TR model). However, the CAL model does account for thermodynamic dissolution so that CaCO3 dissolution rates increase with decreasing pH (Kvale et al 2015a).

**Minor comments: page 2 line 3 : The sentence "... marine carbon export production is well-established Balch (recently summarized by 2018)" should be rewritten as : ".. marine carbon export production is well-established (recently summarized by Balch, 2018)"**

The sentence is changed.

**page 7 lines 2 and 3: " ... suboxic (defined here as below 5 mmol O2)"; did you mean 5 mmol/m3 O2 ?**

The units are fixed.

**pages 18 and 20: The labels on axis of figures 1 and 3 are rather small, please enlarge.**

The labels are enlarged.

**Page 12 lines 18-21: In the sentence starting with: "The MIXED model shows a slight decline in NPP ..." refers to figure A2, which shows plankton biomass but no NPP. Please correct this mismatch or rephrase the sentence.**

The sentence is rephrased.

**Reference:**
**Bach, L.T., et al. A unifying concept of coccolithophore sensitivity to changing car- bonate chemistry embedded in an eco- logical framework. Prog. Oceanogr. (2015), http://dx.doi.org/10.1016/j.pocean.2015.04.012**

---

## Author Response (AR1)

The authors would like to thank the editor for carefully assessing our manuscript. We offer the following responses to the points given:

Figure 1: why is the timeline in the left panel (difficult to read but simulation time goes up to 1400 year?) longer than for the other two panels.

There were several sets of simulations run out to 1400 years and not all plotting scripts were correctly limiting the time series to 500 years for the manuscript. This was corrected in the revised version of the manuscript.

p.5 line 26: What is meant by "stratification starts to reduce"? If it refers to the RU simulation, shouldn't stratification actually increase?

Surface stratification is a transient feature in the WARMING (formerly RU) simulation. It peaks around the time CO2 forcing stabilizes, and continues to diminish as overturning starts to recover. The drivers of this are not clear from our simulations and this is something the authors intend to continue looking in to. The sentence is expanded to reflect this:
Around year 170, near-surface POC export starts to increase in models that do not include ballasting as near-surface stratification starts to reduce (stratification being a transient feature in WARMING that diminishes near the end of the simulation, not shown)...

p.11, lines 14-15: Why would cell size and calcification increase with increasing temperatures? Laboratory studies shows mostly a decrease in calcification. It seems that the authors refer to changes in an evolutionary scale (rise of new species) which would not apply to the time-scale of the simulations presented here.

The sentence is deleted.

Figure A1: please specify that the data presented correspond to the transient response of RD and RU runs in NPP and RESP and temperature. In particular, I assume that the temperatures shown correspond to the response of global ocean average temperature from start to end of each simulation?

The caption is edited to read:

[revised manuscript text omitted]